# Wood Colour Variations of *Quercus* Species in Romania

**Aureliu-Florin Hălălișan** [1] , **Florin Dinulică** [1] , **Dan Marian Gurean** [2],* , **Codrin Codrean** [2] , **Nikolay Neykov** [3] , **Petar Antov** [4] and **Nikolai Bardarov** [4]

1. Department of Forest Engineering, Forest Management Planning and Terrestrial Measurements, Faculty of Silviculture and Forest Engineering, Transilvania University of Brasov, Sirul Beethoven 1, 500123 Brașov, Romania
2. Department of Silviculture, Faculty of Silviculture and Forest Engineering, Transilvania University of Brasov, Sirul Beethoven 1, 500123 Brașov, Romania
3. Faculty of Business Management, University of Forestry, 1797 Sofia, Bulgaria
4. Faculty of Forest Industry, University of Forestry, 1797 Sofia, Bulgaria
* Correspondence: dangurean@unitbv.ro

**Abstract:** Wood colour metrics are increasingly being used in wood technology and ecology studies. Researchers usually determine the colours of the wood after treatment or in different habitats. There is very little research dedicated to the problem of colour variations among one specific species harvested in different forests or regions. The main purpose of the current research is to reveal and estimate the colour variability of oak species. For this study, a total of 89 samples were taken from the heartwood of seven oak species (*Quercus robur* L., *Quercus cerris* L., *Quercus rubra* L., *Quercus pedunculiflora* K. Koch., *Quercus pubescens* Willd., *Quercus petraea* (Matt.) Liebl., and *Quercus palustris* Muenchh.). The CIELAB system was used for the assessment of the colour differences. To determine the colour groups and variations, K-means clustering was used. The results show that colour variations do exist. According to the cluster analysis, at least five types of oak wood can be distinguished (because in some clusters, very few samples were present) in the investigated forests. The differences are mainly observed in terms of the lightness ($L^*$) and yellowness ($b^*$). Redness is not a feature by which oak wood differs, but the wood can be brighter or more yellow in some of the samples. The density of the Romanian oaks in the sample does not influence the colour coordinates. The only coordinate affected is $a^*$, but with a very small probability.

**Keywords:** wood colour; Quercus; CIELAB; quality; Romania; forestry





## 1. Introduction

Superior wood utilisation involves knowing all of the quality characteristics of the wood and how they might affect the wood quality. Wood quality is a relatively new concept [1] and is defined by specific measurable attributes that make wooden materials suitable for one purpose or another. Wood is a material that depends on many factors and shows growth irregularities [2]. In the supply chain, the grading of roundwood is important for the next stage of processing. In practice, wood grading is conducted visually or with basic instruments used to measure any defects, and more objective methods for these processes are desirable. To be productive, grading rules were developed to group wood materials with the same characteristics and ensure that one grade represents the same characteristics and purpose [1]. The grading process is important for the valuation of wood and assures buyers that the materials have the correct characteristics for their needs. Roundwood classification provides the first point of reference for producers on the future utilisation and value of a material. The attribute of colour is important for industry and some enterprises that use hydrothermal treatments such as wood steaming, which is considered a modern way of modifying the colour or tone of the wood [3,4].

Colour is an important characteristic for the aesthetics of wood products. Wood colour differs widely among species as well as within a single tree [5,6], being the result of many

factors. Colour is an attribute that will determine whether the wood can be used for the furniture and decorative veneer industry [7], especially when cumulated with the shape and dimensions [8]. In addition, the wine barrel industry requires wood free of defects, with uniform tree rings in the heartwood and a uniform colour.

In the case of oak species, the heartwood area is the part of most interest in industry. The secondary wood or secondary xylem is so named because it is the result of the growing season activity of a secondary meristem (the cambium) [9]. The wood is structured in annual rings within the heartwood (old rings, physiologically inactive, usually darker in colour) and the sapwood (more recent rings in which the physiological activity is preserved) [9,10].

The heartwood contains hydrolysable tannins (ellagitannins and gallotannins) [11]. In the case of the *Quercus* genus, two monomeric ellagitannin types have been isolated from the heartwood and represent between 40% and 60% by weight of the ellagitannins vescalagin and castalagin [12,13]. Here, ellagitannins constitute the greatest proportion by weight of heartwood extractives [14]. Research [15,16] indicates that the colour of the heartwood is correlated with ellagitannin concentrations and other parameters such as wood structure, anatomy, and genetics, which may also affect it. The soluble parts of ellagitannins, especially the polymeric forms, are the main components of a yellow–brown natural wooden colour [17].

Wood colours have been studied in many contexts. Among the most accurate and commonly used systems for measuring wood colour is CIELAB [18]. The CIELAB system can be described as a uniform colour scale [19]. The system has been used for *Tectona grandis* colour variations [18,20]. Wang et al. [21] examined the textures of solid wood panels and classified their colours through CIELAB. Nguyen et al. [22] predicted the colour changes in larch (*Larix gmelinii*) and poplar (*Populus alba*), defining the coordinates $L*$, $a*$, and $b*$ and estimating their changes after thermal treatment. Mitsui [23] investigated light-irradiated wood with heat treatment using CIELAB colour coordinates. Moya et al. [24] measured the relationship between the chemical structural features of two tree species and the CIELAB coordinates of their colours. Gierlinger et al. [25] managed to reveal the strong relationship between the a* parameter of larch heartwood and its resistance to decay. The mechanical properties of the samples could be predicted by the colour parameters, as was studied in Nasir et al. [26]. Dzurenda et al. [27] examined the changes of maple wood (*Acer pseudopatanus* L.) after steaming with a saturated steam–air mixture. They used the CIELAB system and a difference indicator of the colour changes. The CIELAB method was also used by Dinulică et al. [28] to evaluate the chromatic individuality of fir (*Abies alba* Mill.) compression wood.

Generally, the vast majority of the research considers the colour variations in oak species to be the result of some type of treatment. Barcík et al. [29] estimated colour changes through CIELAB after the thermal treatment of *Quercus robur*. Thermal treatment is among the main reasons researchers seek to measure oak or beech wood colour changes through CIELAB (see [30–34]). There are some papers that examined colour variations as a result of the spatial distribution of the trees. Moya and Alvarado [20] evaluated the environmental factors for wood colour in *Tectona grandis*. Sotelo et al. [34] studied the variation in wood colour among natural populations of *Alanites aegyptiaca* (L.) Delile, *Combretum glutinosum* Perr. ex DC., *Guiera senegalensis* J.F. Gmel., *Piliostigma reticulatum* (DC.) Hochst., and *Ziziphus mauritiana* Lam. trees. The research found that the CIELAB coordinates of the wood colours of these five species vary according to many environmental factors including geographical coordinates.

The main purpose of the current study is to advance the knowledge of the colour variations in the heartwood (part of the secondary xylem) of *Quercus* species from Romania. The study can represent a basis for the selection of wood materials for superior use. Knowledge of the oak wood colour will help both processors and buyers to make decisions about whether a material is suitable for the furniture industry.

## 2. Materials and Methods

### 2.1. Wood Sample Preparation

For this study, 89 samples were taken from the heartwood of seven oak species (*Quercus robur, Quercus cerris, Quercus rubra, Quercus pedunculiflora, Quercus pubescens, Quercus petraea*, and *Quercus palustris*). The first five species are indigenous to Romania and two (*Q. palustris and Q. rubra*) are exotic. The wood samples were taken from Hemeiuș Dendrological Park (Bacău county, Romania) in 1981. The samples were naturally dried over the 1980–2022 period. All of the wood samples measured 6″ × 3″ × 1/2″ IWCS in size (or 15.24 × 7.62 × 1.27 cm) and were part of the Romanian Xiloteque from the Faculty of Silviculture and Forest Engineering (Brașov, Romania). All samples were collected from mature trees during the harvesting process.

In order to be able to carry out the measurements for wood colour determination and density, several steps were followed. In the first step, the samples were processed by sanding using a sander with a grain band that progressively increased from 60× to 120×.

In the second step, the previously sanded samples were cleaned. To clean them, a vacuum cleaner was used as a first method to remove macroscopic particles, followed by wiping the samples with a wet cloth to remove microscopic particles, and finally, a dry cloth was used to avoid leaving water on the samples. The same procedure was used for all samples, and the samples were measured under the same conditions.

### 2.2. Wood Colour Measurements

A Minolta Croma-Meter CR-300 was used to evaluate the wood colour with the same reading parameters as a D65 standard illuminant, a standard observer of $10°$, and a scan window of 6 mm, according to the methodology used by Bekhta and Niemz [35]. The colour of each wood sample represented an average of five measurements (taken from the corners and centre of the wood samples). The reflectance information over the visible spectrum (from 400 to 700 nm) was converted into the CIELAB measuring system [36] with $L^*$, $a^*$, and $b^*$ characteristics. $L^*$ represents brightness, ranging from 0 (black) to 100 (white), while $a^*$ and $b^*$ show the chromaticity coordinates. The $a^*$ axis is in tones from green ($-a$) to red ($+a$) and $b^*$ is in tones of blue ($-b$) to yellow ($+b$). From the values of $L^*$, $a^*$, and $b^*$, the differences in the colour parameters $\Delta L^*$, $\Delta a^*$, and $\Delta b^*$ were used to calculate the colour difference ($\Delta E$). This parameter was obtained in accordance with the methodology used by Bekhta and Niemz [35] and Barcík et al. [29]:

$$\Delta E^* = \sqrt{\Delta L^{*2} + \Delta a^{*2} + \Delta b^{*2}}, \tag{1}$$

where $\Delta E^*$ is the total change in colours.

### 2.3. Wood Density Measurements

For all samples, the density of the air-dried wood was determined using the dimensions 15.24 × 7.62 × 1.27 cm for the volume and the measurements on weight. The weight was determined using a laboratory balance under the same temperature conditions.

### 2.4. Statistical Analysis

The following main methods are used in the present study:

- Cluster analysis. The preliminary data used in the empirical study comprise seven oak species. The cluster analysis includes seven clusters in the k–means clustering algorithm. If each sample falls into a particular cluster, this indicates that colour variations exist between the samples. If certain samples from the same species fall into several clusters, this means that there are premises for intra-species colour variations. All of the statistically significant differences between the clusters are analysed using an index for colour differences ($\Delta E^*$).

- Regression models. The current research also tests the regression models between the parameters of the CIELAB system [18,37] and their relation to the density of each

type of oak, in a similar way to Todaro et al. [19] and Todorovic and Popovic [38]. The models these authors used are simple linear regression models with a single variable.

- Kruskal–Wallis ANOVA test for statistical differences between groups species. The Stata 16 software package (Stata Corp LLC) was used for all calculations. All of the differences are estimated for significance through standard *t*-tests for the differences between two samples after ensuring normal distribution with the Shapiro–Wilk test for normality.

Comparing individual crossbreeds makes sense if an average colour pattern has been created for oak species in the research. This average-based model (average cluster) comprises all of the *L\**, *a\**, and *b\** features of the Romanian oaks. The statistically significant difference between this single model and a separate cluster reveals the individuality of the oak species included in the cluster. Model (1) is transformed for this purpose in the following way:

$$\Delta E^* = \sqrt{\left(L_i^* - \overline{L^*}\right)^2 + \left(a_i^* - \overline{a^*}\right)^2 + \left(b_i^* - \overline{b^*}\right)^2}, \tag{2}$$

where $L_i^*$, $a_i^*$, and $b_i^*$ are the parameters of the cluster $i$ and $\overline{L^*}$, $\overline{a^*}$, and $\overline{b^*}$ are the average parameters of the Romanian oaks in the samples.

If any of the oak species fall into several clusters, the second modification of model (1) can examine the differences:

$$\Delta E^* = \sqrt{\left(L_i^* - L_j^*\right)^2 + \left(a_i^* - a_j^*\right)^2 + \left(b_i^* - b_j^*\right)^2}, \tag{3}$$

where $L_i^*$, $a_i^*$, and $b_i^*$ are the parameters of the particular oak species in the cluster $i$, and $L_j^*$, $a_j^*$, and $b_j^*$ are the parameters of the same oak species in the cluster $j$.

### 2.5. Limitation of the Research

The main purpose of the current study is to advance the knowledge of the colour variations in the heartwood (part of the secondary xylem) of Quercus species from Romania. The first major limitation of the study is related to the size and number of measurements. For this research, only 89 samples were used from the heartwood of seven oak species (*Quercus robur*, *Quercus cerris*, *Quercus rubra*, *Quercus pedunculiflora*, *Quercus pubescens*, *Quercus petraea*, and *Quercus palustris*). For each sample, five measurements were made to establish the colour of the wood samples. With this in mind, any conclusions drawn, as well as future research, must consider the number of samples used. Another major aspect is related to the additional information of the wood samples. No information on the location of the trees is available, since the samples are part of the old Romanian Xiloteque from the Faculty of Silviculture and Forest Engineering (Brașov, Romania). Thus, no correlation was used, since the purpose of the study was to evaluate the variations and to group the species. In addition, the long period during which the wood dried (1980–2022) may have had some influence on the colour, and this must be taken into consideration when interpreting the results.

## 3. Results and Discussion

### 3.1. Colour Characteristics of Oak Species

The lightness (*L\**) of the oak samples ranged from 56.61 (*Q. pedunculiflora*) to 73.62 (*Q. palustris*) (Figure 1). The *L\** varied the most in *Q. pedunculiflora*, from 56.61 to 66. The oak species with the highest level of *L\** was *Q. palustris* (Figure 1).

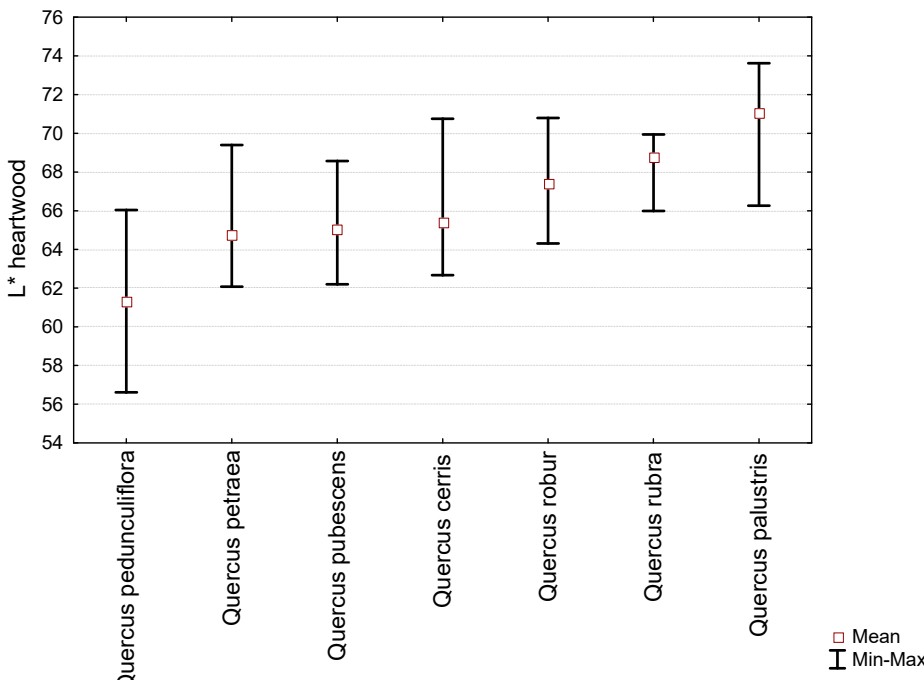

**Figure 1.** Lightness of oak species.

In terms of red colour (positive value of the *a\** parameter), *Q. cerris* had the highest mean (8.32, Figure 2). *Q. palustris* had the lowest value of the *a\** parameter, with a low tone of red. In the case of *Q. robur*, there was a high variation of the *a\** parameter, from 4.82 to 7.33 (Figure 2) with 0.82 standard deviation. Moreover, *Q. cerris* registered a high variation of the *a\** parameter, from 7.03 to 9.07. A low variation of the *a\** parameter was registered in the case of *Q. pedunculiflora*, *Q. petraea*, *Q. pubescens*, and *Q. palustris* (Figure 2).

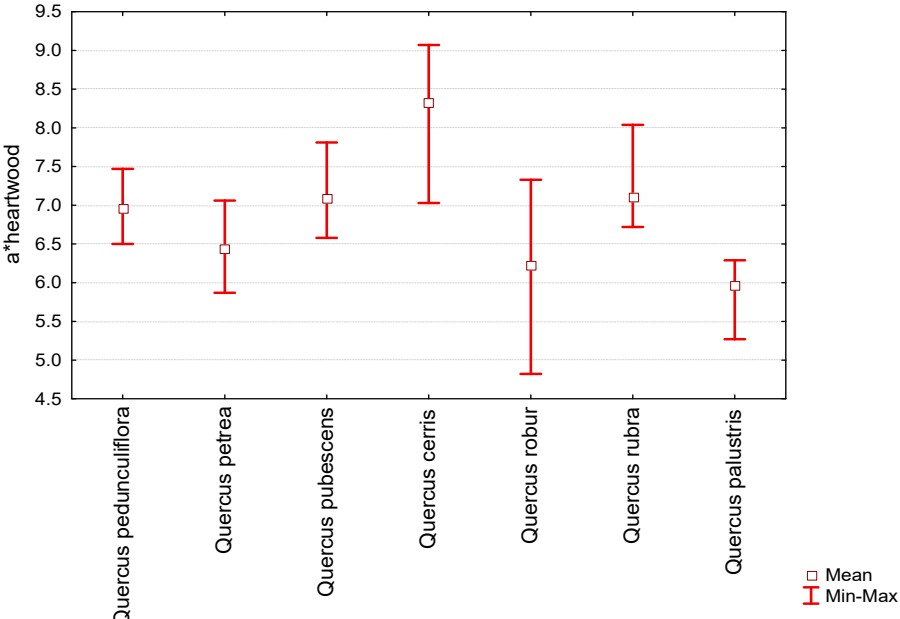

**Figure 2.** Redness (+*a\**) of oak species.

The *b\** parameter refers to the variation from blue (*b\** with negative value) to yellow (b with positive value). In terms of the oak species in this study, the *b\** factor ranged from 20.36 (*Q. palustris*) to 27.89 (*Q. pubescens*) (Figure 3). Only in the case of *Q. robur* was the

variation of the *b\** parameter high (from 20.68 to 26.77 with 1.96 standard deviation). A low variation of the *b* parameter was registered in the case of *Q. pedunculiflora, Q. petraea, Q. pubescens*, and *Q. cerris* (Figure 3).

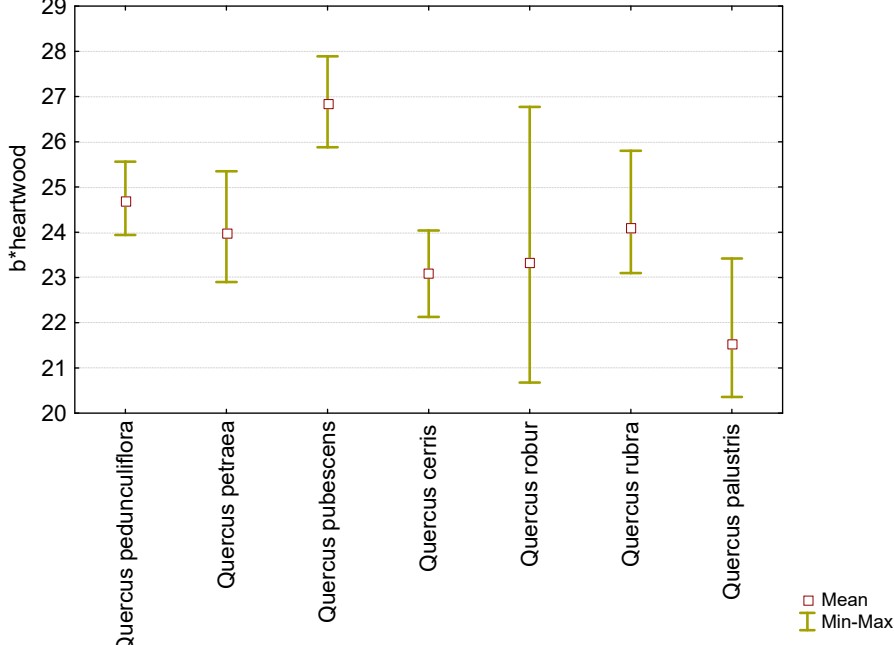

**Figure 3.** Yellowness (+b\*) of oak species.

### 3.2. Grouping Species by Colour Parameters

Cluster analysis can be used as an exploratory data analysis approach to group a set of objects (in this case, wood samples) in such a way that the objects in the same group (called a cluster) are more similar (in some sense) to each other than to those in other groups (clusters). For this study, the cluster analysis was relevant to the differences in the colour of the wood from different tree species, as well as within the same tree species.

To group the samples (and species) according to the ΔE\*, cluster analysis was used. Using the k-means clustering method, seven clusters were obtained. The presence of a sample in a cluster represents the existence of differences in the colour variations from the ΔE\* point of view. If certain samples (and species) fall into several clusters, this means that there are premises for intra-sample colour variations. The presence of samples from different species in each cluster is presented in Table 1. Each species is present in at least in one cluster.

**Table 1.** Investigated oak wood samples and presence in clusters.

| Species | Cluster Number |
|---|---|
| *Quercus robur* L. | 1,3,6 |
| *Quercus palustris* Muenchh. | 1,2,3,5 |
| *Quercus cerris* L. | 1,3,7 |
| *Quercus rubra* L. | 3,6 |
| *Quercus pedunculiflora* K. Koch | 4,6,7 |
| *Quercus pubescens* Willd. | 4,6 |
| *Quercus petraea* (Matt.) Liebl. | 3,6,7 |

Table 1 reveals that there is variety in the participation of each oak type to the number of clusters. Some oak species participated in more than three clusters, while others participated in only two. This suggests a variety of colours within the same species. *Quercus robur* and *Quercus pubescens* were present in two of the clusters. The variations of the colours in

this context were fewer, i.e., the species' colours are resilient. Conversely, *Quercus palustris* had the highest number of variations in colours among the species range. The total number of entities in the samples was 89. In cluster one, there were fourteen entities; in cluster two, there were three; in cluster three, there were twenty-three; in cluster four, there were eleven, in cluster five, there were two; in cluster six, there were fifteen; and in cluster seven, there were twenty-three. A comparison of each cluster with the average coordinates of all of the clusters are presented in Figure 4.

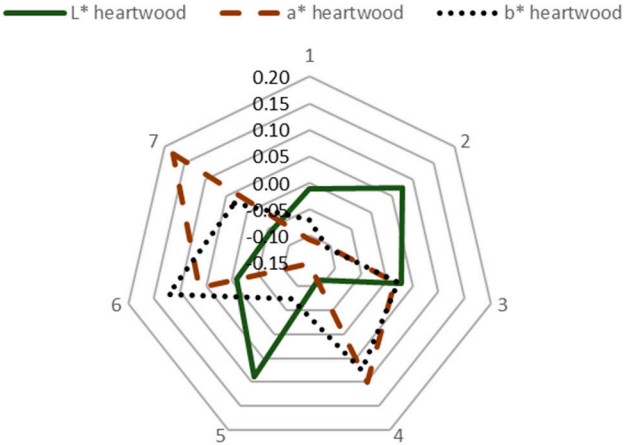

**Figure 4.** Percentage differences between average values of all clusters and cluster coordinates.

The graph shows that clusters five and two had the lightest wood. Clusters four and seven were the darkest. In clusters two and five, only *Quercus palustris* (pin or Spanish oak) was present. This was the lightest wood in the sample. Cluster three included most of the species and was amongst the lightest. Cluster seven represented the reddest wood. The lightest clusters (five and two) included species with lightwood, and the a* and *b** coordinates have the lowest values. Clusters with many samples included seven different samples, and three had different profiles. Cluster seven contained samples with the reddest wood, but was darker than cluster three. Cluster three had almost average values for all of the indicators.

The degrees of changes between the average cluster coordinates and each of the seven clusters discovered are presented in Table 2, according to the classification of Cividini et al. [39].

**Table 2.** Overall changes between average coordinates (average cluster) and coordinates of each of the seven estimated clusters through model (2).

| Cluster Number | ΔE | | L* | a* | b* | Colour Aspect Using L*, a*, b* Parameters |
|:---:|:---:|:---:|:---:|:---:|:---:|:---:|
| 1 | 1.8 | Small difference | 66.47 | 5.93 | 21.62 | |
| 2 | 5.5 | Colour change visible with medium quality screen | 72.29 | 5.96 | 20.82 | |
| 3 | 1.86 | Small difference | 69.07 | 6.66 | 23.60 | |
| 4 | 7.70 | Significant colour change | 59.77 | 7.16 | 24.78 | |

**Table 2.** *Cont.*

| Cluster Number | ΔE | | L* | a* | b* | Colour Aspect Using L*, a*, b* Parameters |
|---|---|---|---|---|---|---|
| 5 | 6.20 | Significant colour change | 73.21 | 5.69 | 21.54 | |
| 6 | 2.76 | Colour change visible with high quality screen | 66.60 | 6.93 | 25.79 | |
| 7 | 3.98 | Colour change visible with medium quality screen | 63.49 | 7.62 | 23.79 | |
| | Average | | 67.27 | 6.56 | 23.13 | |

The table shows that most of the differences were insignificant. There was a significant difference between the middle cluster and clusters four and five. Cluster four includes a 90% sample of *Q. pedunculiflora*. This species was also present in clusters six and seven. In cluster six, the number of samples was one; thus, the analysis was conducted for clusters four and seven, for which the number of samples was five. The Shapiro–Wilk test results for normality in terms of *p*-values are presented in Table 3.

**Table 3.** Shapiro–Wilk tests for normality for each type of coordinate in clusters 4 and 7; α = 0.05.

| Coordinate | Prob > z |
|---|---|
| L* cluster 4 | 0.47 |
| a* cluster 4 | 0.95 |
| b* cluster 4 | 0.29 |
| L* cluster 7 | 0.56 |
| a* cluster 7 | 0.21 |
| b* cluster 7 | 0.68 |

The results show that all of the variables were normally distributed and the *t*-test for differences could be applied. Table 4 presents the results.

**Table 4.** *P*-values for differences between samples in cluster 4 and cluster 7; α = 0.05 and colour differences calculated by model (2).

| | P, *t*-Test, Inequal Variances | *Quercus pedunculiflora* Percentage Difference |
|---|---|---|
| L* | 0 | 7%, cl7 > cl4 |
| a* | 0.112 | - |
| b* | 0.047 | 2%, cl7 > cl4 |
| ΔE * | 4.39 colour difference visible with medium qualityscreen | |

The results reveal that the lightness (*L**) and the yellowness (*b**) were significantly different in cluster seven compared with cluster four. The redness (*a**) was not different among the clusters. In cluster seven, the samples of *Quercus pedunculiflora* were more yellow and were lighter. There are at least two significantly different types of wood texture of *Quercus pedunculiflora* in Romania.

A comparison between the samples of the same species among the clusters was possible for every cluster that contained many samples and for species with many samples spread throughout different clusters. *Quercus robur* was one of these species, after *Quercus pedunculiflora*. It participated in clusters one, three, and six. In the current study, seven samples of the species were analysed in each cluster for each type of coordinate. The test

for normality was conducted for eight samples from each of the clusters one, three, and six, and the results are presented in Table 5.

**Table 5.** Shapiro–Wilk test for normality for each type of coordinate in clusters 1, 3, and 7 for *Quercus robur*; $\alpha$ = 0.05.

| Parameter | Cluster Number | Prob > z |
|:---:|:---:|:---:|
| $L^*$ | 1 | 0.975 |
| | 3 | 0.471 |
| | 6 | 0.312 |
| $a^*$ | 1 | 0.686 |
| | 3 | 0.920 |
| | 6 | 0.059 |
| $b^*$ | 1 | 0.431 |
| | 3 | 0.762 |
| | 6 | 0.977 |

According to the results in the table, all of the variables were normally distributed and the *t*-test could be used for the differences. The results revealed that all of the differences were statistically significant. For lightness ($L^*$), the *p*-values were as follows:

- For the differences between clusters one and three, $p = 0.000$.
- For the differences between cluster one and cluster six, $p = 0.016$.
- For the differences between cluster three and cluster six, $p = 0.003$.

The lightness was statistically different in each cluster. The redness was significantly different between clusters one and six, and between three and six. For the yellowness, all differences were significant. After the estimation and determination of the significance of the differences, it is clear that three colours of wood exist in *Quercus robur*. The degree of differences are as follows, using model (3):

- $\Delta E^*$ cluster one–cluster three = 3.86.
- $\Delta E^*$ cluster one–cluster six = 4.59.
- $\Delta E^*$ cluster three–cluster six = 3.22.

The results reveal that there are differences in the colours, but according to Cividini et al. [39] and Barcík et al. [29], the colour changes are only visible using filters. In other words, there are no different colours, but there are three statistically significant variances of the colours in *Quercus robur*. The greatest difference is between cluster one and cluster six. The difference in $L^*$ is only 2%, but in $a^*$ and $b^*$, the difference is 17%. This means that the common oak in Romania has variations with more red and yellow wood, but not lighter wood. Clusters one and three exhibited slight differences in lightness and yellowness of approximately 5% to 9%.The results confirm the relationship between the individual parameters. Unlike the results of other authors such as Moya and Berrocal [18] for different tree species (*Tectona grandis*), the current results show a significant relationship between $L^*$ and $b^*$. The relationship between $a^*$ and $b^*$ is characteristically positive. The redness of the Romanian oak wood corresponds with its blueness; a 1% increase in the redness would lead to 0.5% of the blueness. For lighter Romanian oaks, their yellowness increases by 0.15% for every $L^*$ percent increase; for each 1% increase, the redness of the Romanian oak heartwood reduces by 0.1% (Figure 5).

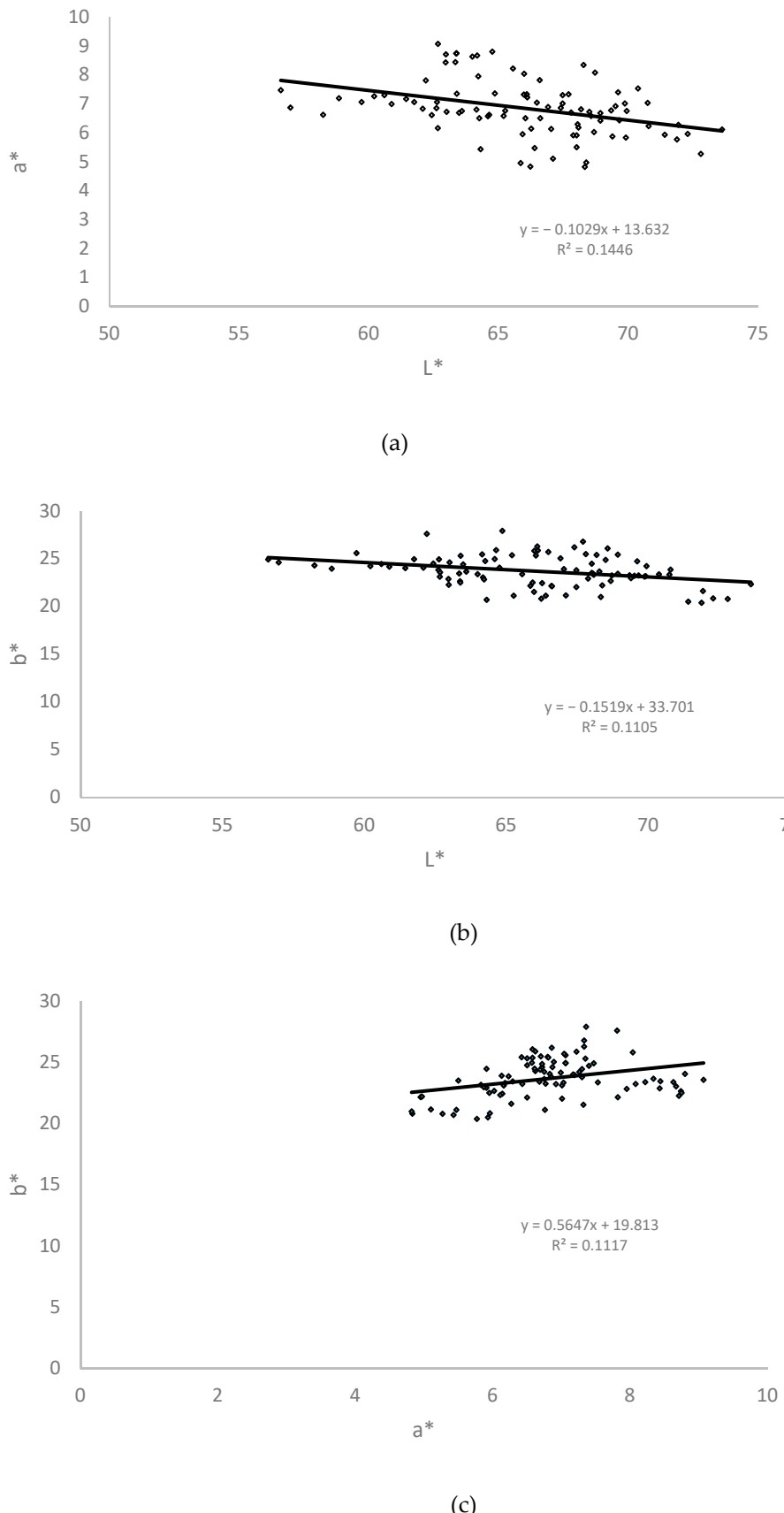

(a)

(b)

(c)

**Figure 5.** Simple regression analyses for the CIELAB parameter relationships: for model (**a**) *p* = 0.000, for model (**b**) *p* = 0.001, and for model (**c**) *p* = 0.001, under α = 0.05.

The wood density of the oak species also influences the colour coordinates. The results of the linear multiple regression are presented in Table 6.

**Table 6.** Regression results for testing the relationship between wood density and colour coordinates; $\alpha = 0.05$.

| Parameter | Coefficient | Density P | R |
|---|---|---|---|
| *L** | 0.003 | 0.079 | 0.070 |
| *a** | 0.018 | 0.020 | |
| *b** | 0.002 | 0.598 | |

The table shows a very weak relationship between the density of the wood species and the colour coordinate system, as well as the CIELAB coordinates. The only statistically significant relationship, although very weak, was between the density of the heartwood and its redness. In only 7% of the cases was the higher density of the oak related to a redder heartwood.

## 4. Conclusions

The colour of oak is a characteristic that has always been used empirically when choosing wood for decorative uses. When colorimetric techniques first appeared, colour become a metric characteristic for grading the quality of wood for specific uses. The present study found that there are peculiarities in the oak species of the Romanian forests. In this way, future research is supported in terms of the extraction and processing of wood from these forests. Although barely distinguishable, the differences certainly differentiate the oak wood species in the researched Romanian forests. According to the cluster analysis, at least five types of oak wood can be distinguished (because in some clusters there were very few samples present) in the investigated forests. The differences are mainly observable in the lightness *L** and yellowness *b**. Redness is not a feature by which oak wood differs, although the wood can be brighter or more yellow in some of the samples. For the majority of the samples of *Quercus robur* and *Quercus pubescens*, the wood cannot be expected to be lighter with more yellow present.

The density of the Romanian oaks in the sample does not influence the colour coordinates. Only the *a** coordinate is affected, but with a very small probability.

**Author Contributions:** Conceptualisation, A.-F.H. and F.D.; methodology, A.-F.H., D.M.G., F.D., and P.A.; validation, F.D., D.M.G., and P.A; formal analysis, N.N. and N.B.; investigation, A.-F.H. and C.C.; writing—original draft preparation, A.-F.H. and. N.N.; writing—review and editing, A.-F.H., F.D., D.M.G., C.C., and N.N.; supervision, P.A.; funding acquisition, D.M.G. and C.C. All authors have read and agreed to the published version of the manuscript.

**Funding:** This research was funded by Transilvania University of Brasov.

**Data Availability Statement:** Not applicable.

**Acknowledgments:** The authors would like to sincerely thank the anonymous reviewers for their valuable insights.

**Conflicts of Interest:** The authors declare no conflict of interest.

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
