# Peer review of "Wood Colour Variations of Quercus Species in Romania"

_forests, doi:10.3390/f14020230_

Round 1

Reviewer 1 Report

The research presented is certainly relevant to the area with which it deals. However, it has some methodological shortcomings. For example, among the methods, the CIELAB numerical colour evaluation system is adequately presented, even with a detailed description of the individual colour components, while the method for calculating the colour changes is not explained at all. In general, of course, the equation for calculating the colour differences is known. It is calculated from the color differences of the colour components between the two compared surfaces. However, it is not clear here how these colour changes were calculated for individual tree species in the first place, or what the authors used as a reference colour from which the colours of the tree species differed. As a result, the reader loses the comprehensibility and significance of the results presented when reading the results. The parameters of the colour measurement are also not fully presented (lines 123 to 128). There is a lack of information on the geometry of the measurement, the type of light and information on whether or not the specular component was included in the measurement. Wood belongs to the inhomogeneous materials. For this reason, only 5 measurements per sample are not sufficient.
Another methodological problem lies in the preparation of the samples (lines 106 to 117). It is not clear from the description whether all samples are from the Xyloteca or only samples from the last two tree species. If research is concerned with differences in the colour of oak wood in Romania, then the authors should at least provide a little information about the growing sites of the trees and their approximate location. It is also not clear when the trees were cut and how long and at what humidity the wood was dried. If they really are such old wood samples, then the evaluation of their colour is highly questionable. Despite the fact that the surfaces have been sanded, this is still a problem because during such a long resting/drying period of the wood, there are various migrations of wood tannins from the interior to the surface, which significantly affects the colour of the wood, and the colour of such wood is certainly not a representative wood colour of currently growing trees. Tree cutting and sampling should be done systematically at different locations and over approximately the same time period, all wood should be dried by the same procedure, and samples should be conditioned to their constant weight under the same climatic conditions before measurement, since the moisture content of the wood has a significant effect on colour. Not to mention the wetting of the surface of the wood used in the study (line 120).
The study also does not state how the density of the wood was determined and what density the study is talking about, most likely it is the density of air-dried wood.
The results should additionally explain how the same cluster analysis can be used to draw conclusions about differences in the colour of wood from different tree species as well as within the same tree species.
The article also needs some technical corrections. For example, in some places the language is less understandable or needs revision. For example, lines 20, 45, lines 56 through 62... The use of passive voice is often inappropriate. The double consecutive use of the plural in title is questionable (Colours Variations”. Colour Variations of Quercus Wood Species in Romania?
The CIELab system is not named consistently. CIELab is used once, CIE L* a* b* the second time.
Colour parameters could be written in italics, for example, L*, a*, b*, and deltaE*.
The metric system should be used (line 112).
Latin names of tree species are written in italics, even in pictures.
Colour coordinate values do not have units. The L* value must not be given in percent (line 129, 153...).

Author Response

Thank you for your effort to review our paper!

In the attached file we address all issues!

Reviewer 2 Report

The work presents the evaluation of the colour of oak heartwood specimens taken from seven different oak species. The research subject is well-reasoned, as the colour may serve as an effective feature for detecting the used oak species. In the same way, the producers may profit from knowledge about which species to use for attaining the required colour hue. From this viewpoint, the issue is very relevant.

However, I must state two serious objections concerning the work. The first is for the methods, especially for the experimental material sampling. The research mentioned in the paragraph above requires an absolutely novel approach to the study material selection. It is necessary to specify several variables such as the number of stems belonging to the given oak species, site conditions for the sampling localities, and specimen-taking layout across the stem. Such research has a considerable extent, puts high demands, and needs up to several years to accomplish. This does not facilitate an instant publication of the results.  I may declare firmly, based on my own experience with studying beech wood properties. The results obtained by my own were subsequently generalised in the context of the whole land. Your results do not seem to allow generalisation, and some declarations would profit from pondering them much deeper.

The second one is for the Introduction. This section is very extensive and diverse, much outside the issue discussed in the paper. I do not understand why there is debated wood discolouration induced by thermal treatment land laser treatment, as this is completely out of the frame of this paper. This literature would be well-reasoned to if the paper topic were thermal colour homogenisation, for example, in the case of oak wood originating from different oak species. I insist on re-writing the Introduction.

.

Author Response

Thank you for effort to review our paper!

In the attached file we adress all issue.

Round 2

Reviewer 2 Report

You did not satisfy me with your answers to my questions. But OK, I accept your opinions as authors.

Happy New Year

Author Response

Dear Reviewer,

Thank you for your support and for your attention.
We made all changes and we update the references list as follow:
We full remove the reference 9 and 10 since the reviewer indicate that is not relevant for our study:
9. Kačík, F., Kubovský, I. Chemical changes of beech wood due to CO2 laser irradiaton. J. Photochem. Photobiol. A: Chemistry 2011, 222(1): 105-110.
10. Németh, R., Ot, Á., Takáts, P., Bak, M. The effect of moisture content and drying temperature on the colour of two poplars and Robinia wood. BioResources 2013, 8(2), pp.2074-2083.
And the reference number 8 (Moya, R., Berrocal, A. Wood colour variation in sapwood and heartwood of young trees of Tectona grandis and its rela?onship with plantation characteristcs, site, and decay resistance. Ann. For. Sci. 2010) was move from Introducton to Methods and Results chapter since is an important reference for results ( article strong connected to our subject).We add two more relevant references:
4.Sedliačiková, M., Moresová, M., 2022. Are Consumers Interested in Colored Beech Wood and Furniture Products?. Forests 2022, 13 (9), p.1470.
26.Nasir, V., Fathi, H., Fallah, A., Kazemirad, S., Sassani, F., Antov, P. Prediction of mechanical properties of artificially weathered wood by color change and machine learning. Materials 2021, 14(21), p.6314.

Thank you!

Happy New Year!
